# Dietary Nutrient Intake and Blood Micronutrient Status of Children with Crohn’s Disease Compared with Their Shared-Home Environment, Healthy Siblings

**DOI:** 10.3390/nu14163425

**Published:** 2022-08-20

**Authors:** Stephanie Brown, Catherine L. Wall, Chris Frampton, Richard B. Gearry, Andrew S. Day

**Affiliations:** 1Department of Pediatrics, University of Otago, Christchurch 8011, New Zealand; 2Department of Medicine, University of Otago, Christchurch 8011, New Zealand

**Keywords:** Crohn’s disease, nutritional status, pediatrics, micronutrients, inflammatory bowel disease, diet, oral nutrition supplement, partial enteral nutrition

## Abstract

(1) The nutritional status of children with Crohn’s disease (CD) is rarely described. This study aimed to assess the dietary intake and blood micronutrient status of children with CD compared with their healthy, shared-environment siblings. (2) Methods: This observational study included children with CD (cases) and their shared-environment siblings (controls). The dietary nutrient intake was assessed with a four-day food/beverage diary and was compared with the recommended daily intakes (RDI). Blood micronutrient concentrations were measured using laboratory methods. The nutritional analyses were completed through a multivariate analysis of variance between groups. Between-group comparisons of single-nutrients were assessed using a Mann–Whitney U-test. Chi-squared analyses compared the proportion of children who did not meet the RDI for each nutrient. The results were significant at 0.05. (3) Results: The dietary intake was similar for most nutrients, except the controls had a lower intake of vitamins A and E, copper, zinc, iron, and selenium (*p* < 0.05). Children using partial enteral nutrition had significantly higher intakes of many micronutrients. It was common for both groups to not meet the RDI’s—more than 50% of cases for 9 nutrients and more than 50% of controls for 13 nutrients. (4) Conclusion: New Zealand children with CD and their shared-environment siblings did not meet the RDI for several micronutrients. Dietary education and/or micronutrient supplementation may be required.

## 1. Introduction

Crohn’s disease (CD), a subtype of inflammatory bowel disease (IBD), features active and chronic inflammation in the gastrointestinal tract. Although the pathogenesis of IBD is unclear, it is represented by a dysregulated host immune response to intestinal bacteria in an individual who is genetically susceptible, along with responses to the external environment [1]. Symptoms of CD may include abdominal pain, weight loss, malnutrition, joint pain, and diarrhea. In children, ongoing symptoms can directly impact upon growth, development, and quality of life [2]. Given that both CD and ulcerative colitis (UC), another major type of IBD, are considered incurable, management options focus upon establishing control of the inflammation and ongoing maintenance of remission over time. 

Diet is closely linked to IBD, as it is associated with both an increased and decreased risk of developing IBD [3,4,5,6] and is can be used as a form of therapy [7]. Approaches such as exclusive enteral nutrition (EEN) are especially relevant in children and adolescents during times of rapid growth and developmental change when nutrient requirements are increased. EEN can reverse inflammatory changes in the gut, which leads to an improvement in symptoms, such as diarrhea and abdominal pain [8]. The provision of adequate micronutrients and macronutrients during EEN also improves the nutritional status and anthropometric markers [9]. 

In addition to EEN, the use of which aims to induce remission, ongoing dietary interventions may have roles in the maintenance of remission. These include the potential role of exclusion diets (where potential triggers are excluded) or diets that enhance key protective/beneficial elements [10,11,12]. The inflammatory changes seen in CD can disrupt normal absorption and lead to nutrient deficiencies [13,14]. As well as having roles in correcting malnutrition, nutritional interventions may also have an impact on pro-inflammatory responses within the body.

Various dietary interventions have been proposed to manage IBD, especially to prevent disease symptoms [13]. Such dietary interventions include the exclusion of lactose-containing dairy products or changes in fiber content [13,14,15]. Given the role of diet in the development and management of IBD, diet is a common area of concern for patients. However, there remain many gaps in our understanding of the impacts of diet and dietary interventions on both the disease process and the short- and long-term nutritional status. There is strong interest among patients to explore dietary management strategies to prevent and manage IBD. However, data on the use of diet as a treatment option for IBD remain inconclusive or limited. 

The primary aims of this study were to assess the dietary intake of ambulatory children with CD compared with the recommended daily intakes (RDIs) and dietary intake of healthy, shared-environment siblings. A secondary aim was to assess the blood micronutrient status of both cohorts of children.

## 2. Materials and Methods

### 2.1. Participants

This observational, cross-sectional study recruited children with CD and their healthy siblings between the ages of 5 to 17 years living within the Canterbury region of New Zealand (2017–2019). The CD group (cases) comprised children with CD who had a confirmed diagnosis of CD at least 4 months prior to recruitment based on the Porto criteria for diagnosis [16]. Cases were included if they were eating a “normal” habitual diet, and the use of PEN was also permitted. The healthy sibling group (controls) resided in the same household as their sibling with CD, so as to control for food exposure and environment. 

Exclusion criteria for children with CD included those currently receiving treatment with EEN, parenteral nutrition support, or those taking any nutritional supplements, as this would not reflect a “normal” diet and would prevent an accurate assessment of their habitual dietary and nutrient intake. Any participants with other underlying chronic medical conditions or comorbidities, including obesity, diabetes, or celiac disease, which might impact on habitual intake, were excluded. Siblings who resided between split households were excluded due to lack of a shared food environment.

Verbal and written information was provided to children prior to participation. Consent was obtained for children < 10 years of age by the child’s parent or guardian, and those over 10 years of age provided informed consent in the presence of their parent or guardian to participate. The study was approved by the University of Otago Health Ethics Committee, reference H17/168.

### 2.2. Data Collection

#### 2.2.1. Participant Characteristics 

The characteristics of the cases were collected from clinical files on the hospital patient records system. These included demographics, disease location at the most recent assessment, medical therapies, number of hospitalizations, prior surgical interventions, age, self-reported ethnicity, sex, and the number of years since disease diagnosis. Age, gender, and anthropometry were collected from the controls.

#### 2.2.2. Disease Activity

Disease activity at the time of recruitment was determined by the clinical team using both the Physician Global Assessment (PGA) [17] and the Pediatric Crohn’s Disease Activity Index (PCDAI) [18]. The PGA provided an overall assessment of disease activity using a 0 to 3 scale (0 = nonactive disease; 1 = mild disease activity; 2 = moderate disease activity; 3 = severe disease activity). The PCDAI provided a specific assessment of the disease activity, measured with a 0 to 100 scale (<10 indicating remission, 100 indicating the most severe grade of activity).

#### 2.2.3. Anthropometry 

All participants and their parents were asked to report the participant’s height and weight at the time of survey completion. Each child’s body mass index (BMI) (kg/m^2^) was calculated and converted to Z scores using the World Health Organization (WHO) anthropometry cut-off values [19]. Nutritional risk was defined using the WHO BMI Z-score index for 5 to 19 years of age: “healthy weight/low risk”>−1 to <+1 SD, “thinness/moderate risk” <−2 SD, and “severe thin-ness/high risk” <−3 SD. “Overweight” >+1 SD to <+2 SD, and “obese/high risk” >+2 SD.

### 2.3. Dietary Intake Assessment

#### Four-Day Food and Beverage Diary

A four-day food and beverage diary was selected for its accurate assessment of intake [20,21] and collected the dietary intake on three non-consecutive weekdays and one weekend day. The food and beverage diary scored a Microsoft Word readability score of 10.3 years. Therefore, participants aged ten and under were advised to complete the food and beverage diary with the help of their parent/guardian. 

Participants recorded brand names for foods wherever possible and were asked to collect the food label information/wrappers for any unusual foods and ready meals consumed to help identify or clarify items. For homemade dishes, participants were asked on a separate page in the diary the individual ingredients and quantities for the whole dish, along with a brief description of the cooking method and how much of the dish they had consumed. Participants were asked to consider leftovers when recording how much they consumed.

After each day, participants recorded if their intake was typical for that day (and if not, the reason why) and details of any of the dietary supplements taken. The diary also contained a series of questions about usual eating habits (for example, type of milk or fat spread usually consumed) to facilitate coding in cases where details were omitted in the diet record.

### 2.4. Blood Collection

Blood samples from children with CD and their healthy siblings were taken at the time of the child’s IBD clinic visit. A total blood volume of 10 mL (using standard pediatric blood vials) was taken by a trained pediatric nurse.

Blood samples were sent to Canterbury Health Laboratory (CHL) immediately after collection for analysis. The following were measured: sodium, urea, creatinine, phosphate, magnesium, full blood count, liver function tests, calcium, glucose, vitamin A, vitamin E, vitamin D, iron studies (including ferritin, iron, and transferrin saturation), vitamin C, albumin, zinc, selenium, copper, erythrocyte sedimentation rate (ESR), c-reactive protein (CRP), folate, and vitamin B12. The results were interpreted using the New Zealand National Pediatric Reference range values [22] and converted into z-scores to enable comparative analyses. 

### 2.5. Analysis of Dietary Intake

The four-day food and beverage diaries were analyzed using [23] a New Zealand/Australian-specific nutritional software package (Foodworks, Xyris Software, Foodfiles version 10). All of the food and beverage diaries were double entered into the Foodworks software. The first data entry was completed by a registered dietitian (study coordinator) and the second data entry was completed by a dietetic assistant. Any data discrepancy was noted and clarified directly with the participant if possible, to minimize reporting errors. The mean intake of the macronutrients and micronutrients across the three days was calculated as the mean (± standard deviation). The mean intake was compared with the RDIs of The New Zealand Pediatric Recommended Daily Intake dietary reference values [24]. 

### 2.6. Statistical Analysis

All of the outcome measures were analyzed cross-sectionally and temporally using SPSS software version 26 (IBM Corp, Armonk, NY, USA). Descriptive statistics were calculated for all of the outcome parameters for each cohort according to normality of distribution and were reported as the mean (standard deviation) and/or number (percent). A linear random-effects mixed-model was conducted to evaluate the cross-sectional and temporal associations of outcome measures. Nutritional analyses were completed by multivariate analysis of variance to determine which nutrients were driving the difference in intake between groups. Between-group comparisons of single nutrient intakes for continuous data were assessed using the Mann–Whitney U test. 

Chi-squared analyses were used to compare the proportion of children who did not meet the RDI for each nutrient between groups. Correlations between nutritional profile, blood results, disease location, PCDAI, and clinical parameters were measured using a Spearman correlation coefficient test. The results were considered statistically significant at the 0.05 level.

## 3. Results

### 3.1. Study Participants

Sixty children and adolescents aged 5 to 17 years were recruited. Thirty-three children with CD (cases) and twenty-seven healthy, shared-environment siblings (controls) were included (Table 1). There were more male cases recruited and more females controls. The mean BMI of the cases was lower than that of the controls (*p* = 0.008). The average time since the diagnosis of Crohn’s disease was 3.1 years.

### 3.2. Dietary Intake

All of the children provided complete four-day food and beverage diaries for analysis. The mean dietary intake of children with CD and their siblings was similar for most nutrients (Table 2), except the intake of vitamins A and E, copper, zinc, iron, and selenium (*p* < 0.05) were significantly lower for the siblings. The dietary intake of 20 vitamins and minerals were compared with the NZ pediatric RDIs. It was common for both children with CD and their siblings to not meet their RDI’s for age; more than 50% of children with CD did not meet the RDI for 9 nutrients and more than 50% of siblings did not meet RDI for 13 nutrients (Table 2). The nutrient RDIs that were not met differed between groups. 

### 3.3. Partial Enteral Nutrition and Nutrient Intake

Almost half of the children with Crohn’s disease (42%) used PEN with a micronutrient dense oral nutritional supplement. The nutrient intake of the children who recorded oral nutritional supplement use in their food and beverage diaries was compared with children who did not consume oral nutrition supplements (Table 3). The nutrient intake of children using PEN was significantly higher for polyunsaturated fats, vitamin A equivalents, thiamine, riboflavin, niacin, folate, vitamin C, copper, iodine, iron, and selenium (*p* < 0.05) compared with the children not using PEN. Only dietary fiber (*p* = 0.023) and vitamin K (*p* = 0.035) were lower in the children using PEN compared with the children not using PEN.

A post hoc analysis compared the nutrient intake of the children with CD who were not using PEN with their siblings. There were no significant differences in nutrient intake between children not taking PEN compared with their healthy siblings, except for polyunsaturated fats (*p* = 0.027) (data not shown).

### 3.4. Blood Micronutrient Status 

Blood samples were collected for 30 (91%) cases and 27 (100%) controls. Table 4 contains the results of the blood micronutrient results. This table includes data on 1 SD below the reference mean. The cases had significantly lower blood micronutrient concentrations for copper, folate, and zinc (*p* < 0.05), while the controls had a significantly lower blood micronutrient status for phosphate (*p* < 0.05). Only three (10%) cases had elevated CRP > 5 mg/L, and the controls had a significantly lower CRP compared with the cases (*p* < 0.05). The albumin concentration was adequate in both groups.

### 3.5. Multivariate Analyses 

The multivariate analyses found no significant correlations between dietary nutrient intake and PCDAI, disease location, or blood nutrients within or between groups (data not shown).

The Spearman’s correlation coefficient test found significant correlations between PCDAI scores and disease location for iron, copper, and magnesium. The PCDAI scores were negatively significantly associated with a lower serum iron concentration (*p* < 0.05). The presence of ileal disease (L1) and colonic disease (L2) according to the Paris Classification was correlated with significantly higher copper and magnesium concentrations (*p* < 0.05).

## 4. Discussion

Nutrition plays an important role in both the pathogenesis and treatment of CD in children. Those with active CD experience anorexia, early satiety, impaired absorption, and enhanced intestinal transit that, together, impair the overall nutritional status and commonly impact adversely upon normal growth and development in this crucial life-stage [25]. Nutritional status has been identified as the main determinant of both surgical and clinical outcomes for patients with CD [26]. As a result, the correction of nutritional deficiencies plays a crucial role in disease treatment as pharmacological strategies. 

This is the first study to analyze the dietary intakes and serum nutritional status of children with quiescent CD compared with their shared-home environment, healthy siblings. Children with CD who did not use PEN had a similar nutrient intakes as their shared-home environment siblings. However, children with CD who were using PEN had a significantly higher nutrient intakes compared with those children with CD who were not using PEN. Both cohorts of children, including those who used PEN, did not meet the RDIs for several nutrients. Lastly, there were no significant correlations between dietary nutrient intake and blood nutrient concentrations. These findings differ from prior studies comparing the diet of children with IBD to a healthy control group.

The present study showed no significant differences in nutrient intakes, except for PUFAs, between children with CD not taking PEN compared with their healthy, shared-environment siblings. These finding are likely due to the participants’ shared food-environment. We previously showed that children with CD had poorer food-related quality of life (FRQoL) than their healthy, shared-home environment siblings and a healthy control group [27]. However, the siblings also had poorer FRQoL compared with the healthy control group, indicating that there may be an adverse psychological impact on family members who share a food-environment with someone with a chronic bowel disease. A study by Chuong et al. [28] demonstrated that families altered their food-shopping list and meal-planning routine to accommodate their child with IBD. Consequently, the nutrient intake of siblings living within the same household will likely be affected. 

The long-term use of oral nutritional supplements (partial enteral nutrition) improves nutritional status in children with IBD [9], and is recommended to assist in the maintenance of disease remission [29,30]. Although there is evidence to support the use of PEN, this strategy is likely under-utilized, possibly because of poor adherence. Our study illustrated that children with CD who use daily PEN in addition to a normal diet have significantly better nutritional intakes compared with children with CD who do not use PEN for polyunsaturated fats, vitamin A equivalents, thiamine, niacin, folate, vitamin C, vitamin D3, vitamin E, copper, iodine, iron, selenium, and zinc (*p* < 0.05). In addition to an improved nutrient intake, the use of PEN has been associated with better anthropometry indices, improved endoscopic endpoints, reduced mucosal cytokine levels, and has a positive effect on maintaining remission, leading to significantly fewer relapses after one year [31]. Taken together, it appears that dietary intake alone may be insufficient to meet the nutritional requirements of children with CD and the use of PEN will likely improve nutritional and maybe disease status.

All of the study participants did not meet the RDI for their age for several micronutrients, particularly calcium and potassium. Inadequate dietary calcium and potassium intakes in children with IBD have been reported previously [32,33]. This present study showed that three quarters of both children with CD and their siblings did not consume the recommended intake of calcium. Adequate calcium and potassium intake, as well as optimal blood vitamin D status, during childhood and adolescence is essential to achieve peak bone-mineral density (BMD) [34,35]. This is particularly important for children with CD who are at increased risk of diminished (BMD) due to chronic systemic inflammation, malnutrition, corticosteroid use, delayed puberty, growth retardation, low BMI, and an inactive lifestyle [36,37]. The early identification of patients at risk of low BMD, including those with an inadequate calcium intake, is essential, as early dietary education or nutritional supplementation to achieve the RDI will help prevent osteoporotic fractures later in life [38]. 

The observed fiber deficiencies in the current study are consistent with previous studies [39,40,41,42]. In addition, a further study suggested that a low fiber diet is used by patients as a preventative strategy to avoid disease flare-ups [43]. However, a recent systematic review including 16 studies (only one involving children) demonstrated that fiber-avoidance is associated with an increased risk of a CD flare [44]. Interestingly, many of the polymeric formulas used by children for PEN and EEN are fiber-free, which may be counterproductive in the prevention of disease activity management, or to prolong periods of remission. Currently, research in this area is examining the effect of fiber-containing feeds (used as EEN therapy) [45] and the effects of a high-fiber diet in people with UC [46] which will help to identify the impact of fiber on a gut with active IBD.

The recent ESPGHAN guidelines do not recommend fat-soluble vitamin supplementation of vitamins A, K, and E in the absence of liver disease. However, they do recommend vitamin D supplementation when serum levels fall below 50 nmol/L [47]. In the present study, the RDIs for fat-soluble vitamins (A, D, E, and K) were not being met by either study cohort. Although this study only included children with quiescent CD, a study by Bousvaros et al. [48] showed a higher prevalence of vitamin E and A deficiencies among children with more active disease. Conversely, a prospective study by Costa et al. [49] showed that both children with active and inactive CD had significantly lower serum vitamin A and E concentrations compared with a healthy control group of children. Deficiencies of vitamins K and E in children with IBD are not widely reported in the literature; however, one cross-sectional study of 63 children with CD showed a 54% prevalence of vitamin K deficiency, which was correlated with disease activity [50]. 

Regarding iron, it is standard clinical practice to monitor iron indices and to replace iron in the presence of iron-deficiency anemia (IDA) [21], as IDA is indicative of intestinal losses and active disease. Iron deficiency is common in pediatric IBD [49,50,51,52] and more than a quarter of our IBD cases were iron deficient compared with only 11% of siblings. Despite a higher prevalence if IDA among the cases, a higher proportion of siblings (56%) did not meet the RDI for iron compared with 39% of cases. Adequate iron intake has been reported previously. Hartman et al. [42] reported that children with IBD had dietary intakes of iron 112% (*p* < 0.05) above the recommended daily amount. 

Zinc deficiency is less commonly assessed in clinical practice or reported in the literature. In this study, a third of cases were zinc deficient. Conversely, while no siblings showed zinc deficiency, a third were not meeting the RDI for zinc. In IBD, adequate zinc intake and blood status is important, as research shows that adequate zinc levels reduce pro-inflammatory cytokines and maintain intestinal permeability in Crohn’s disease [53]. Zinc blood status should be interpreted cautiously, given that approximately 70% of plasma zinc is bound to albumin, and therefore zinc measurements could be largely adjusted by the albumin concentrations [54]. With that being said, no children with CD had low blood albumin concentrations in this study.

Folate deficiency is frequently detected in patients with IBD, especially those with CD [55]. The dietary intake of vitamin B9-rich foods may be adequate or higher than in healthy subjects; however, the absorption of folate may be adversely impacted by disease, given that active inflammation leads to a higher demand for folic acid, which is associated with an increased production of granulocytes and inflammatory cells [56]. This corresponds to the finding in the present study with a higher dietary intake of folate among cases compared with the controls; however, there was also a higher prevalence of serum folate deficiency in the cases compared with the controls (18.2% vs. 7.4% (*p* < 0.05)). Interestingly, Costa et al. [49] reported normal serum folate concentrations in all of their study participants.

Selenium, copper, magnesium, and biotin were all inadequately consumed through diet. Biotin dietary intakes and serum levels have not been previously reported. Biotin is a B vitamin that is a cofactor that catalyzes critical steps in the metabolism of amino acids, fatty acids, and glucose. It also plays key roles in histone modifications, gene regulation (by modifying the activity of transcription factors), and cell signaling [57]. However, it’s specific role in CD has not previously been reported. While a poor intake of magnesium has been observed in a prior study [42], copper intake was sufficient [42], and serum deficiencies of selenium and copper are notably more common in pediatric IBD [58,59]. This is especially true in New Zealand, where the soil selenium levels are reportedly low, with wide-geographical variation in selenium soil levels around the country [60]. With that being said, this present study showed no correlation between dietary intake and micronutrient blood levels.

The limitations of the current study include its cross-sectional design and the relatively small sample size (33 cases and 27 controls). However, this study’s strengths include the use both dietary and serum analyses of nutritional status, and this is the first study to compare the dietary intakes and serum nutritional status between children with CD in remission and their shared-home environment siblings. Furthermore, this study’s use of a four-day food and beverage diary to assess the energy and nutrient intake is another key strength. Previous studies that reported poorer nutrient intakes compared with healthy controls [28,29,30,31,32,33] used retrospective food recall and food-frequency questionnaires, which are less reliable and have a lower specificity. The ESPGHAN Nutrition in Pediatric IBD position paper recommends a three to five-day dietary record as the most suitable method to quantitatively assess energy and nutrient intake [34].

## 5. Conclusions

We have shown that children with CD and their shared-home environment siblings do not meet their dietary requirements for several vitamins and micronutrients. We have also shown the impact that chronic disease may have on other family members regarding food choice and dietary sufficiency. Given that children taking PEN had a significantly better nutritional status compared with the children not taking PEN, PEN may be an advantageous long-term strategy to enhance nutritional status. Routine dietary analysis should be completed by a pediatric dietitian to ensure children with CD are meeting the RDI recommendations, as well as the provision of dietary counselling to improve the nutrient intake accordingly. The findings of this present study suggest that nutritional supplementation may need to be considered if children cannot meet the micronutrient RDI for their age from food alone. Further research is required to develop effective strategies that utilize both food and supplements (including PEN) in order to ensure an adequate nutrient intake of children with CD. 

## Figures and Tables

**Table 1 nutrients-14-03425-t001:** Participant characteristics and demographics.

	Crohn Disease (n = 33) Mean [SD] or n [%]	Siblings (n = 27) Mean [SD] or n [%]
Age, years	13.15 (6.36)	13.2 (7.78)
Gender		
Male	16 (49)	9 (33)
Weight z-score	0.39 (1.33)	0.88 (1.33)
Height z-score	0.56 (1.06)	0.47 (1.63)
BMI, kg/m^2^ z-score	−0.09 (1.32)	0.76 (0.97)
Nutritional risk		
Low risk	27 (82)	25 (93)
Medium risk	4 (12)	2 (7)
High risk	2 (6)	0 (0)
Disease location (Montreal Classification)		
L1 (terminal ileum)	15 (45)	-
L2 (colon)	7 (21)	-
L3 (Ileocolon)	22 (66.7)	-
L4 (upper GI)	6 (18)	-
P (perianal involvement)	8 (24)	
Years since diagnoses [range, years]	3.09 (1–6)	-
PGA score		
0 Remission	21 (64)	-
1 Mild	6 (18)	-
2 Moderate	4 (12)	-
3 Severe	2 (6)	-
PCDAI score		
<10 Remission	21 (64)	-
10–25 Mild	6 (18)	-
25–40 Moderate	4 (12)	-
>40 Severe	2 (6)	-
Concurrent therapies		
Infliximab	6 (18)	-
Corticosteroids	0 (0)	-
Thiopurines	11 (33)	-
Maintenance enteral nutrition only	14 (42)	-
Other		
Prior resection	2 (6)	-
>1 disease flare since 2018	11 (33)	-

BMI, body mass index using the WHO standard indices; nutritional risk as defined by the WHO BMI z-score aged 5–19 years; PGA, physicians global assessment scale; PCDAI, pediatric Crohn’s disease activity index; SD, standard deviation; WHO, World Health Organization.

**Table 2 nutrients-14-03425-t002:** Dietary nutrient intakes of cases compared with the controls.

Nutrients	IBD N = 33 Mean (SD)	Sibling N = 27 Mean (SD)	IBD (Cases), < RDI ^#^ for Age N (%)	Sibling (Controls), < RDI ^#^ for Age N (%)	*p* Values
Fibre (g)	19.19 (8.17)	20.81 (6.62)	18 (55)	19 (70)	0.367
Potassium (mg)	2550.67 (891.84)	2573.20 (1004.58)	20 (61)	19 (70)	0.915
Phosphate (mg)	1201.14 (478.46)	1116.89 (326.08)	18 (55)	17 (63)	0.490
Calcium (mg)	731.86 (328.76)	696.93 (279.31)	24 (73)	20 (74)	0.691
Iron (mg)	14.08 (6.96)	10.38 (4.70)	13 (39)	15 (56) *	0.031
Zinc (mg)	11.96 (6.24)	8.92 (3.50)	4 (12)	9 (33) *	0.038
Selenium (µg)	60.02 (25.09)	47.23 (17.15)	15 (45)	18 (67) *	0.049
Vitamin A (µg)	1065.07 (476.68)	842.69 (389.77)	9 (27)	8 (30) *	0.034
Thiamine (mg)	1.55 (0.73)	1.64 (1.12)	5 (15)	8 (30)	0.661
Riboflavin (mg)	2.05 (0.84)	2.02 (0.89)	1 (3)	3 (11)	0.999
Niacin (mg)	19.51 (9.26)	15.27 (5.09)	6 (18)	6 (22)	0.057
Vitamin B6 (mg)	2.77 (1.15)	2.63 (1.87)	1 (3)	2 (7)	0.876
B12 (µg)	3.16 (2.03)	2.76 (1.29)	8 (24)	7 (26)	0.110
Folate (µg)	280.47 (123.54)	227.96 (102.18)	18 (55)	22 (81)	0.415
Vitamin C (mg)	72.02 (54.25)	62.80 (52.14)	10 (30)	10 (37)	0.063
Vitamin D (µg)	5.69 (6.40)	3.54 (3.81)	19 (58)	22 (81)	0.077
Vitamin E (mg)	9.70 (5.24)	6.67 (2.98)	13 (39)	21 (78) *	0.011
Vitamin K (µg)	8.19 (11.10)	8.64 (14.66)	30 (91)	26 (96)	0.978
Copper (mg)	1.45 (0.81)	1.02 (0.33)	15 (45)	18 (67) *	0.016
Magnesium (mg)	269.40 (108.34)	249.29 (78.25)	18 (55)	15 (56)	0.447
Biotin (µg)	6.76 (5.10)	6.61 (3.60)	30 (91)	25 (93)	0.769

**^#^** Dietary nutrient intakes of cases include those taking PEN in addition to food, New Zealand paediatric recommended daily intake dietary reference ranges [24], * *p* < 0.05. Multivariate analysis of variance test, between subjects.

**Table 3 nutrients-14-03425-t003:** Dietary intake of children with Crohn’s disease who consumed and did not consume partial enteral nutrition.

Nutrients	Children with Crohn Disease Habitual Diet ± PEN		
	No PEN (N = 19)	PEN (N = 14)	
	Mean (SD)	Mean (SD)	PEN Higher *p*-Value < 0.05
Energy (KJ)	7532.16 (2911.97)	7942.73 (1623.04)	-
Protein (g)	81.29 (42.78)	76.37 (21.13)	-
Dietary fibre (g)	22.11 (8.84)	15.38 (5.41) *	0.035 (No PEN)
Monounsaturated fat (g)	25.29 (14.30)	25.51 (7.71)	-
Polyunsaturated fat (g)	9.93 (4.93)	16.86 (8.15) *	0.007
Sodium (mg)	2276.52 (770.89)	2343.32 (732.55)	-
Iodine (µg)	64.81 (36.48)	131.58 (65.45) *	0.001
Potassium (mg)	2648.56 (1044.91)	2422.66 (659.57)	-
Phosphorus (mg)	1253.57 (610.69)	1132.58 (216.76)	-
Calcium (mg)	657.83 (371.64)	828.68 (243.17)	-
Iron (mg)	11.54 (5.38)	17.40 (7.58) *	0.019
Zinc (mg)	10.20 (5.91)	14.26 (6.12) *	0.005
Selenium (mg)	49.29 (19.19)	74.04 (25.59) *	0.005
Vitamin A equivalents (µg)	901.89 (384.96)	1278.45 (513.21) *	0.029
Thiamine (mg)	1.21 (0.60)	1.99 (0.67) *	0.002
Riboflavin (mg)	1.78 (0.83)	2.40 (0.75) *	0.043
Niacin (mg)	16.59 (8.15)	23.32 (9.54) *	0.047
Vitamin B6 (mg)	2.65 (1.29)	2.93 (0.96)	-
Vitamin B12 (µg)	3.08 (2.51)	3.26 (1.23)	-
Folate (µg)	232.86 (105.7)	342.73 (120.70) *	0.013
Vitamin C (mg)	42.48 (21.08)	110.66 (60.54) *	0.000
Vitamin D3	4.72 (7.82)	6.96 (3.79) *	0.002
Vitamin E (mg)	7.24 (2.66)	12.93 (6.08) *	0.002
Vitamin K (µg)	11.88 (13.01)	3.36 (5.23)	-
Copper (mg)	1.09 (0.52)	1.93 (0.89) *	0.003
Magnesium (mg)	272.03 (132.16)	265.95 (71.09)	-
Fluoride (mg)	0.01 (0.01)	0.00 (0.00)	-
Manganese (µg)	4153.92 (2443.87)	5171.06 (1655.83)	-
Molybdenum (µg)	35.55 (17.70)	29.11 (13.04)	-
Pantothenic Acid (mg)	1.80 (1.16)	1.19 (0.51)	-
Chromium (µg)	12.48 (10.71)	11.81 (8.90)	-
Biotin (µg)	7.81 (4.90)	5.39 (5.22)	-

* Partial enteral nutrition (PEN) > no PEN (*p* < 0.05) Multivariate analysis of variance.

**Table 4 nutrients-14-03425-t004:** The blood micronutrient status of the study participants.

Nutrient	CD (N = 30) Mean (SD)	Sibling (N = 27) Mean (SD)	CD, 1 SD below the Reference Mean N (%) ^#^	Sibling, 1 SD below the Reference Mean N (%) #
Protein g/L Vitamin A ug/L	74.9 (4.6) 505.7 (941.1)	76.1 (3.7) 367.9 (86.9)	7 (23.3) 9 (30.0)	3 (11.1) 4 (14.8)
Vitamin E μol/L	20.2 (7.0)	22.3 (6.8)	5 (16.6)	5 (18.5)
Vitamin D nmol/L	72.7 (25.1)	71.9 (20.8)	2 (7.0)	4 (14.8)
B12 pmol/L	444.3 (152.8)	463.7 (196.5)	5 (16.6)	4 (14.8)
Folate nmol/L	27.8 (18.3)	25.0 (8.3)	6 (20.0)	2 (7.4) *
Copper μmol/L	16.09 (4.39)	14.97 (2.8)	7 (23.3)	4 (14.8) *
Zinc μmol/L	11.93 (2.09)	12.09 (1.47)	11 (36.6)	0 (0.0) *
Selenium μmol/L	1.15 (0.18)	1.17 (0.28)	2 (7.0)	3 (11.1)
Iron μmol/L	15 (6.7)	17.6 (5.7)	9 (30.0)	3 (11.1)
Potassium mmol/L	3.97 (0.27)	4.15 (0.49)	4 (13.3)	2 (7.4)
Phosphate mmol/L	1.37 (0.23)	1.34 (0.23)	3 (10.0)	5 (18.5) *
Magnesium mmol/L	0.9 (0.1)	0.8 (0.1)	1 (3.0)	1 (3.1)
Sodium mmol/L	139.9 (2.1)	139.5 (1.8)	5 (16.6)	2 (7.4)
*C*-Reactive protein mg/L	2.91 (1.41)	1.07 (0.71)	1 (3.1)	6 (20.0) *

**^#^** Compared with the Paediatric New Zealand national serum laboratory reference guide, **p* < 0.05. CD, Crohn’s disease; SD, standard deviation.

## Data Availability

Not applicable.

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
