# Peer review of "Dietary Nutrient Intake and Blood Micronutrient Status of Children with Crohn’s Disease Compared with Their Shared-Home Environment, Healthy Siblings"

_nutrients, 2022, doi:10.3390/nu14163425_

Round 1
Reviewer 1 Report
In this study, Stephanie Brown, et al. assessed the dietary intake of ambulatory children with Crohn’s Disease (CD) compared with the recommended daily intakes (RDIs) and dietary intake of healthy, shared-environment siblings. They conclude that New Zealand children with CD and their shared-environment siblings were not meeting the RDI for several micronutrients, and dietary education and/or micronutrient supplementation may be required.
Strengths of this study:
Study question is valid.
Adequate literature review was performed.
The author's major findings were clearly presented. They adequately address the stated research objectives.
The research results validate the author's conclusions.
I have a few suggestions to consider:
- Consider including P-values in Tables.
- Authors may want to add if patients were taking any nutritional supplements that could have influenced the results.
- What is the socioeconomic status of the study population?
- Add lab findings in Table 1 or as separate table.
Author Response
Dear reviewer,
Thank you for your valuable feedback and suggestions which will improve the manuscript quality. We address your comments below in a point-by-point fashion.
Comment 1:
Consider including P-values in Tables.
All p-values that were calculated for the blood analyses was completed as p < 0.05. We have included p values for for
Table 2: Dietary nutrient intakes of cases compared to controls lines 194-195.
Table 3: Dietary intake of children with Crohn’s disease who consumed and did not consume partial enteral nutrition Lines 209-210.
Comment 2:
Authors may want to add if patients were taking any nutritional supplements that could have influenced the results.
Amended to include
Lines 76-79
Exclusion criteria for children with CD included those currently receiving treatment with EEN, parenteral nutrition support, or those taking any nutritional supplement as this would not reflect a “normal” diet and prevent accurate assessment of their habitual dietary and nutrient intake.
Comment 3:
What is the socioeconomic status of the study population?
Unfortunately, this data was not collected.
Comment 4:
Add lab findings in Table 1 or as separate table.
Laboratory findings are located in Table 4 - Lines 225-226 titled Blood micronutrient status of study participants
Sincerely
Stephanie Brown
Reviewer 2 Report
This study aim to assess the dietary intake and blood micronutrient status of children with Crohn’s disease compared with their healthy, Pediatric Crohn’s disease is a rare, inflammatory bowel disease characterized by severe, chronic inflammation of the intestinal wall or any portion of the gastrointestinal tract. But the frequency of pediatric Crohn’s disease is increasing; in particular, there has been a recent increase in the incidence of Crohn’s disease in children less than 6 years old Generally, Crohn’s disease is more severe among children and adolescents than in adults. Therefore this study assumes an added value for a growing disease especially in Caucasian regions. This work deepens analytical aspects of the intake of various microelements not always evaluated in the analyzes, arriving at some encouraging conclusions for a correct management of young patients. From an ethical point of view, we remind you to ask for the informed consent of the parents as the study was carried out on patients under age. It also recalls in the conclusions to the line 376-377 a typo of transcription.
Author Response
Dear reviewer,
Thank you for your feedback and suggestions which have improved the manuscript quality. We have addressed your comments below in a point-by-point fashion.
Comment 1
From an ethical point of view, we remind you to ask for the informed consent of the parents as the study was carried out on patients under age.
Thank you for this reminder and we have amended the manuscript to include Lines 83-86
Consent was obtained for children <10 years of age by the child’s parent or guardian, and those over 10 years of age provided informed consent in the presence of their parent or guardian to participate.
Comment 2:
It also recalls in the conclusions to the line 376-377 a typo of transcription.
We have deleted this transcription error.
Sincerely,
Stephanie Brown